# The critical current of disordered superconductors near 0 K

A. Doron [1✉], T. Levinson[1], F. Gorniaczyk[1], I. Tamir[1] & D. Shahar[1]

An increasing current through a superconductor can result in a discontinuous increase in the differential resistance at the critical current. This critical current is typically associated either with breaking of Cooper-pairs or with the onset of collective motion of vortices. Here we measure the current–voltage characteristics of superconducting films at low temperatures and high magnetic fields. Using heat-balance considerations we demonstrate that the current–voltage characteristics are well explained by electron overheating enhanced by the thermal decoupling of the electrons from the host phonons. By solving the heat-balance equation we are able to accurately predict the critical currents in a variety of experimental conditions. The heat-balance approach is universal and applies to diverse situations from critical currents to climate change. One disadvantage of the universality of this approach is its insensitivity to the details of the system, which limits our ability to draw conclusions regarding the initial departure from equilibrium.

---

[1] Department of Condensed Matter Physics, The Weizmann Institute of Science, Rehovot 7610001, Israel. ✉email: adamdoron1@gmail.com

One of the central properties superconductors is their critical current ($I_c$)[1–3] the maximal current ($I$) they are able to maintain. The sudden onset of resistance ($R$) at $I_c$ is usually associated with one of two mechanisms: de-pairing, which occurs when the kinetic energy of a Cooper-pair exceeds its binding energy (the superconducting gap)[4,5], or de-pinning of vortices, when the $I$-induced Lorentz force acting on vortices exceeds their de-pinning force, setting them in motion[4,6,7]. Typically, in type-II superconductors under the application of magnetic field ($B$), de-pinning occurs at lower $I$ rendering the de-pairing current a theoretical upper-bound[4,8].

Due to the practical significance of $I_c$ the bulk of the scientific effort was centered around increasing its value at finite temperatures ($T$'s) rather than on its fundamental, $T = 0$, value. In a recent publication, $I_c$'s of superconducting amorphous indium oxide films (a:InO) have been studied at low $T$'s and high $B$'s near the high critical field of superconductivity, $B_{c2}$ (~13 T)[9]. The authors found that $I_c \sim |B - B_{c2}|^\alpha$, with $\alpha \approx 1.6$ that is close to the mean-field value of 3/2 indicating, as they pointed out, that $I_c$ is a result of the combined action of de-pairing and de-pinning where the increasing $I$ initially suppresses the order parameter (by pair-breaking), helping the Lorentz force to overcome the pinning. While by using this approach they were able to suggest a resolution to the ubiquitous linear $B_{c2}(T)$ as $T \to 0$[10,11], their theory is not yet refined enough to offer a quantitative prediction to the value of $I_c$ itself.

Our purpose in this article is to suggest that a different mechanism, which is Joule self-heating, is behind $I_c$, this mechanism inevitably becomes more dominant as $T \to 0$. Self-heating occurs when the power dissipated by the measurement $I$ exceeds the rate of heat removal from the electrons. To analyze this process we model our experiment as being comprised of four independent subsystems (Fig. 1a) that are thermally coupled via lumped thermal resistors ($\tilde{R}$'s): The electrons, the host a:InO phonons, the substrate phonons and the liquid helium mixture (in which our sample is immersed in our dilution refrigerator). While our system as a whole is driven out of thermal equilibrium by our measurement $I$, it maintains a steady-state where we assume that we can treat each subsystem as being at local equilibrium albeit at different $T$'s represented by $T_{el}$, $T_{ph}$, $T_{sub}$ and $T_0$ as indicated in Fig. 1a.

Our electronic subsystem is thermally linked to its phonons via $\tilde{R}_{el-ph}$, mediated by electron-phonon coupling[12–15]. The a:InO phonons are, in turn, linked to the substrate's phonons via acoustic transfer at the interface between the different solids[16], which transfer their heat to the helium mixture through a thermal-boundary resistance at the interface known as Kapitza resistance, $\tilde{R}_K$[17–19].

Under steady state conditions the power ($P$) flowing across each boundary $\tilde{R}$ is equal to the Joule heating $P \equiv I \cdot V$ delivered to the electronic subsystem. A finite $P$ flowing through the $\tilde{R}$'s results in a $T$-difference between each pair of subsystems. If one of the $\tilde{R}$'s is significantly larger than the others it will constitute a thermal bottleneck, impeding the cooling process, and the largest $T$ difference will develop across it. A straightforward analysis, given in Supplementary Note 6, reveals that the thermal bottleneck is between the electrons and the phonons ($\tilde{R}_{el-ph}$) and henceforth we assume that all other subsystems are in equilibrium with each other (the functional form of Equation (1) is general and also applies to the other $\tilde{R}$'s illustrated in Fig. 1a, therefore we can write the terms of Equation (1) as if $\tilde{R}_{el-ph}$ is the thermal bottleneck and not lose generality). The $T$-differences across $\tilde{R}_{el-ph}$ is determined by a heat-balance equation:

$$P = \Gamma\Omega\left(T_{el}^\beta - T_{ph}^\beta\right) \tag{1}$$

where $\Omega$ is the sample's volume and $\Gamma$ and $\beta$ are parameters characterizing $\tilde{R}_{el-ph}$[12,14,15].

It turns out that Equation (1) can lead to dramatic behavior. If a rise in $T_{el}$ that results from an increase in $I$ causes a sufficiently steep increase in $R(T_{el})$, which is certainly the case in our type-II superconductor (in our experiment $R(T_{el}) \approx R_0 e^{-T_0/T_{el}}$), then below a critical $T_{ph}$ value the heat-balance equation acquires two stable solutions for $T_{el}(I)$: a low $T_{el}$ solution where $T_{el} \gtrsim T_{ph}$ and a high $T_{el}$ solution where $T_{el} \gg T_{ph}$[12,20]. The jump at $I_c$ is simply a manifestation of the system switching discontinuously between the two stable $T_{el}$ solutions, and the sudden increase in $V$ at $I_c$ results from $V = I_c R(\text{low } T_{el} \to \text{high } T_{el})$.

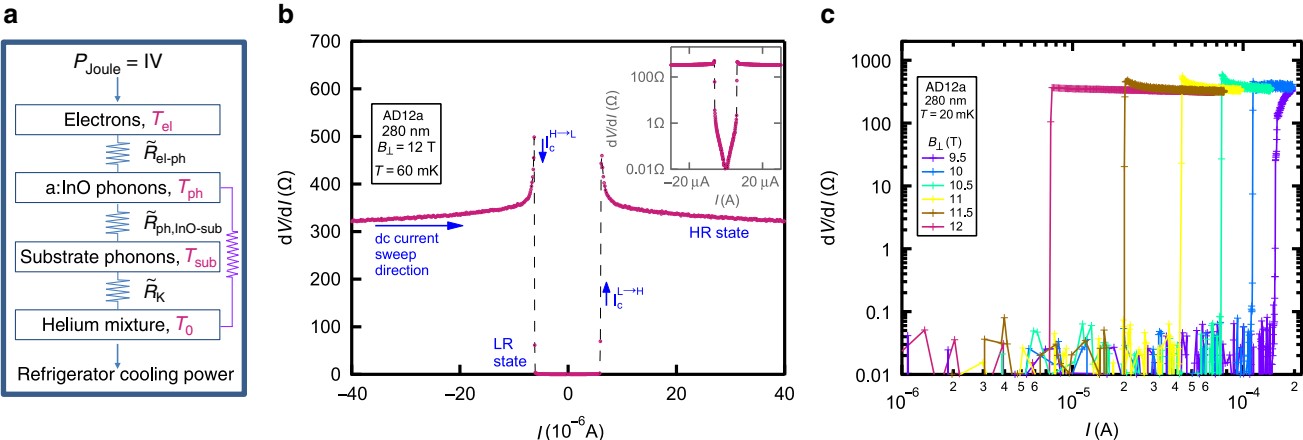

**Fig. 1 Discontinuities in the $I$–$V$'s. a** Schematic diagram of the heat-flow between subsystems. Joule heating $P_{Joule} = I \cdot V$ dissipated in the electrons subsystem by our measurement $I$ flows to the a:InO phonons via a thermal resistor $\tilde{R}_{el-ph}$, then from the a:InO phonons to the substrate phonons via $\tilde{R}_{ph,InO-Sub}$ and into the helium mixture of the dilution refrigerator via $\tilde{R}_K$. The purple $\tilde{R}$ stands for other less significant parallel processes $\tilde{R}$'s (for example, a:InO phonons can transfer heat directly to the liquid helium via Kapitza resistance, we omit this process as the boundary area between the substrate and liquid helium is 20 times larger than that of the a:InO and liquid helium). **b** $dV/dI$ vs. $I$ (linear scale) of the 280 nm film at $T = 60$ mK and $B_\perp = 12$T. The measured $dV/dI$ separates to a LR state and to a HR state. The transition between these states occurs discontinuously at $I_c^{H \to L}$ (marked by a downwards pointing arrow) and at $I_c^{L \to H}$ (marked by an upwards pointing arrow). In the inset we plot the same data on a semi-log scale, this enables us to see the small yet finite $R$ in the LR state. **c** $dV/dI$ vs. $I$ (log–log) at $T = 20$ mK of the 280 nm thick film. The color-coding marks different $B_\perp$'s where blue is $B_\perp = 9.5$T and magenta is $B_\perp = 12$T. At different $B_\perp$'s there are abrupt jumps in $dV/dI$, at $I_c$, which decreases with $B_\perp$.

We present the results of a systematic study of $I_c$ in superconducting a:InO at $0.5 > T > 0.01$ K and $B$'s $12 \geq B \geq 9$ T (where $B_{c2} \simeq 13$T), for samples of various thicknesses in both perpendicular $B$ ($B_\perp$) and in-plane $B$ ($B_\parallel$). We demonstrate that the $V$-jump at $I_c$ results from the behavior expected from the heat-balance Equation (1). Furthermore, the value of $I_c$ can be accurately determined using only measurements done at $I \to 0$ and at $I \gg I_c$. We also show that $I_c$ is not consistent with either de-pairing or de-pinning mechanisms, nor with their combined action.

## Results

**$I_c$ at low $T$ and high $B$.** In Fig. 1b, c we depict several low-$T$ current–voltage characteristics ($I$–$V$'s) typical of our study obtained from the 280 nm sample. In Fig. 1b we plot $dV/dI$ vs. $I$ measured at $B_\perp = 12$ T and at $T = 60$mK. The measured $dV/dI$ clearly separates to two regimes, a low-resistive (LR) state at low $|I|$'s and a high resistive (HR) state at high $|I|$'s. The transition between these states occurs discontinuously at two different $I_c$ value; we define the measured $I$ where the HR $\to$ LR transition occurs as $I_c^{H\to L}$ and the measured $I$ where the LR $\to$ HR transition occurs as $I_c^{L\to H}$ (as marked in the Fig. 1b). Typically our measured $I$–$V$'s did not show a pronounced hysteresis as $I_c^{H\to L} \approx I_c^{L\to H}$ (see "Discussion" section) therefore throughout the manuscript we refer to both as $I_c$. In the inset of Fig. 1b we re-plot the data of the main graph on a semi-logarithmic scale. Plotted this way it can be seen that in the LR state, even at low $I$'s, there is a small but finite $R$. This is because a thin film type-II superconductor in the presence of high disorder and $B$ reaches $R = 0$ only at $T = 0$[6,21]. In Fig. 1c we plot $dV/dI$ vs. $I$ on a semi-logarithmic scale measured at 20mK and at 6 $B$-values between $B = 9.5$–$12$ T ($B_{c2}$ for this sample, plotted in Supplementary Fig. 2, was ~13 T). All curves exhibit a jump of several orders of magnitudes in $dV/dI$ at a well-defined, and $B$-dependent $I_c$: Increasing $B$ towards $B_{c2}$ results in a decrease in $I_c$. Similar $B$ dependence of $I_c$ is observed in all of our samples.

**Heat-balance analysis.** We next demonstrate, using the heat-balance approach (Equation (1)), that the $I$–$V$'s are well described in terms of electron self-heating[12–15,22–24]. Inspecting Equation

(1) we see that the only unknown variable is $T_{el}$, which we need to obtain independently. For that, we assume that all deviations from Ohm's law are due to an increase in $T_{el}$ and not from other non-linear effects (we shall review the flaws of this assumption in the discussion). Under this assumption, we convert the raw $I$–$V$'s obtained from our 280 nm film at $B_\perp = 12$ T at several $T$'s plotted in Fig. 2a to effective $T_{el}$ and plot the results in Fig. 2b[20,23,25] (see Supplementary Note 3 for a detailed description of the heat-balance analysis).

Finally we plot, in Fig. 2c, $P + \Gamma\Omega T_{ph}^\beta$ vs. $\Gamma\Omega T_{el}^\beta$ alongside the fit to Equation (1) (dashed black line), which our data follow for more than 4 decades, and from which we extract the parameters $\beta = 5.1$ and $\Gamma\Omega = 1.48 \times 10^{-5}$ W $\cdot$ K$^{-\beta}$ (the values of $\beta$ for our samples at various $B$'s are given in Supplementary Table 1). The systematic deviations at low $P$'s are addressed in Supplementary Note 7.

**Calculation of $I_c$ from the heat-balance analysis.** Encouraged by the excellent fit of our data to the heat-balance theory, we further provide a quantitative test for its validity by using it to predict the values of $I_c$ for our superconductors. This is achieved by numerically solving Equation (1) to obtain the theoretical lower and upper bounds of the bi-stability $I$-interval, $I_c^{min}$ and $I_c^{max}$, for our $B$ and $T$ range as demonstrated in Fig. 3.

In Fig. 3a we plot the measured $dV/dI$ vs. $I$ for the 280 nm thick sample at $T_{ph} = 60$ mK and $B_\perp = 12$ T. In Fig. 3b we plot both sides of Equation (1), the magenta curve is the right-hand-side ($\Gamma\Omega(T_{el}^\beta - T_{ph}^\beta)$) where $\beta$ and $\Gamma\Omega$ were extracted from the heat-balance analysis (this curve is simulated). The other curves are $I^2R(T_{el})$, the left-hand side of Equation (1) (plotted for four different $I$ values marked in Fig. 3a), where $R$ is the measured $R(T)$ at equilibrium (the dashed portions are a low $T$ extrapolation of the $R(T)$). A valid $T_{el}$ solution is where each of the four Joule-heating curves intersect with the magenta curve. In the inset of Fig. 3b we plot these possible $T_{el}$ solutions as a function of the driving $I$.

At low $I$ ($-3\,\mu$A) Equation (1) has only one solution at $T_{el} \approx 0.06$ K, close to $T_{ph}$. This $T_{el}$ solution is plotted in the inset of Fig. 3b as a thick dashed line.

Increasing $|I|$ to $I = -6.2\,\mu$A brings about a dramatic development. At this $I$ the blue curve intersects the magenta

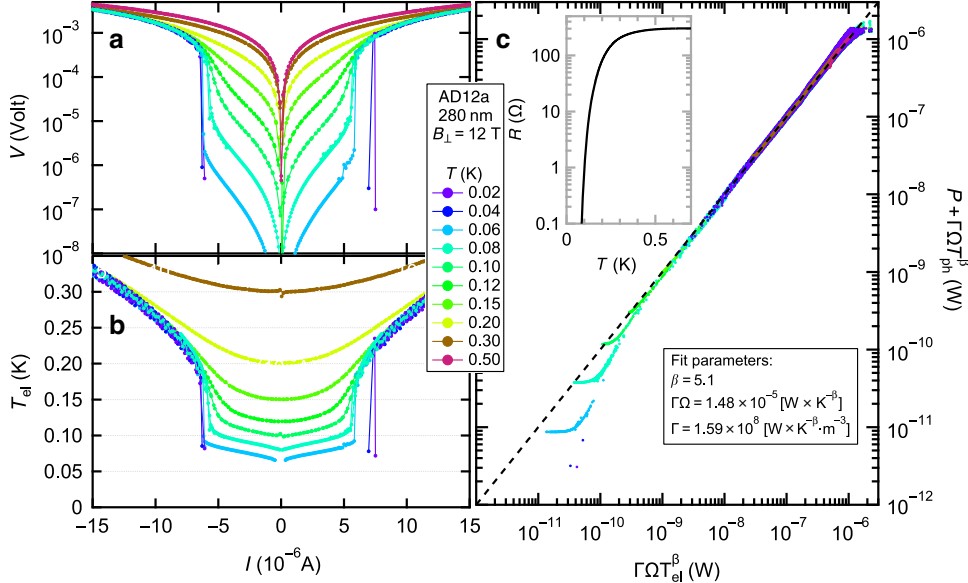

**Fig. 2 Heat-balance analysis. a** $|V|$ vs. $I$ of the 280 nm thick sample at $B_\perp = 12$ T. The color-coding marks different $T$'s. **b** $T_{el}$ vs. $I$ extracted from the data of (**a**) using the zero-bias $R(T)$ (inset of (**c**)) as an electron thermometer. **c** Fitting the data to Equation (1). By collapsing the different isotherms of (**a**) such that $P + \Gamma\Omega T_{ph}^\beta = \Gamma\Omega T_{el}^\beta$ we extract the parameters $\beta$ and $\Gamma\Omega$. See Supplementary Note 3 for additional details.

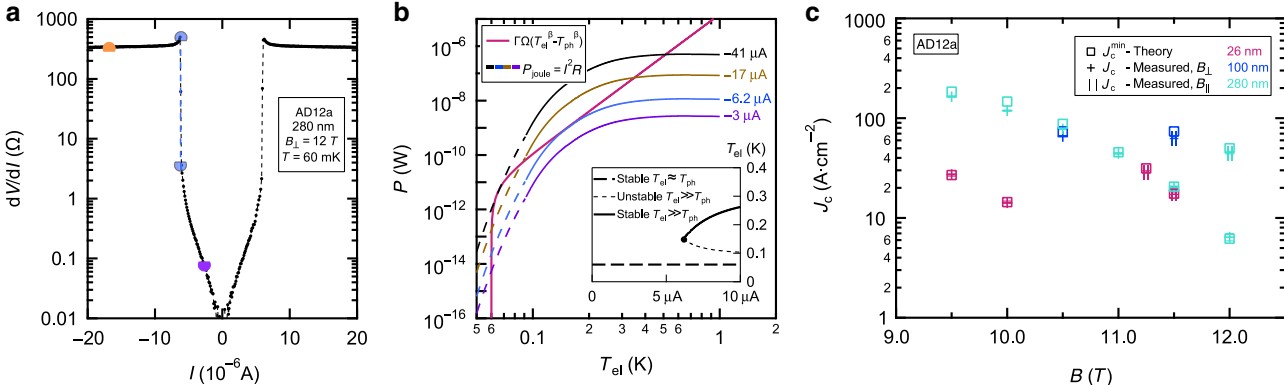

**Fig. 3 Graphical solution of the heat-balance equation. a** $dV/dI$ (log-scale) vs $I$ of the 280 nm thick film at $T_{ph} = 60$ mK and $B_\perp = 12$ T. The purple and brown semicircles mark $dV/dI$ at $I = -3\,\mu$A and $-17\,\mu$A, respectively, the two light blue semicircles connected by a dashed blue line mark $dV/dI$ at $I = -6.2\,\mu$A. **b** Graphical solution of Equation (1) for $T_{ph} = 60$ mK and $B_\perp = 12$ T. The magenta curve marks the right-hand side of Equation (1) vs $T_{el}$, the four additional curves correspond to the left-hand side of Equation (1) (joule heating $P = I^2R$) where the color-coding of the purple, blue and brown curves corresponds to the same $I$ values of (**a**) and the black curve corresponds to $I = -41\,\mu$A. A crossing point between the two sides of Equation (1) occurs at a possible $T_{el}$ solution. The purple, blue, brown, and black curves intersect with the magenta curve once, twice, thrice, and twice respectively. We identify the theoretical boundaries of the bi-stability interval as $[I_c^{min} = 6.2\,\mu$A, $I_c^{max} = 41\,\mu$A] (see text). Inset: The three possible $T_{el}$ solutions vs $I$. For $I < I_c = 6.2\,\mu$A there is a single solution of $T_{el} \approx T_{ph}$ (thick dashed line). For $I > I_c$ Eq. (1) also has a stable (thick continuous line) and an unstable (thin dashed line) elevated $T_{el}$ solutions. **c** A comparison between measured and theoretical $J_c$'s for different samples and $B$ orientations. $J_c$ values extracted from the solution of Equation (1) are marked by squares, crosses and ||'s mark the measured $J_c$ in $B_\perp$ and $B_\parallel$, respectively. The color-coding marks the sample thickness. Error bars are determined by the density of $I$ points in the $I - V$'s.

curve twice as they become tangent at $T_{el} \approx 150$ mK. We identify this $I$ value with $I_c^{min}$, the theoretical lower-bound of the $I$-interval where a thermal bi-stability can exist. The main result of this work is that this theoretical $I_c^{min}$ matches the measured $I_c$. This is demonstrated in Fig. 3a where $I = -6.2\,\mu$A is marked by a dashed blue line that coincides with the measured $I_c^{H\to L}$.

Increasing $|I|$ beyond $|I_c^{min}|$ drives the sample into the bi-stable regime where there are three solutions to Equation (1). For example, at $I = -17\,\mu$A there are three crossing points between the brown and magenta curves in Fig. 3b marking three different $T_{el}$ solutions for Equation (1). The middle solution is an unstable fixed-point and the low and high $T_{el}$ solutions are stable. The unstable $T_{el}$ solution is marked in the inset of Fig. 3b by a thin dashed line and the stable high $T_{el}$ solution is marked by a continuous thick line. The black curve ($I = -41\,\mu$A) corresponds to $I_c^{max}$ where the black and magenta curves intersect twice as they become tangent at $T_{el} \sim 70$ mK. Above $I_c^{max}$ the low-$T_{el}$ stable solution and the unstable high $T_{el}$ solution coincide and vanish leaving the system only with the high-$T_{el}$ stable solution.

In Fig. 3c we plot the theoretical $J_c^{min}$ which is the critical current density corresponding to $I_c^{min}$ (squares) for samples with various thicknesses together with our measured $J_c$ (crosses and ||'s representing $B_\perp$ and $B_\parallel$ respectively where we have used the measured $I_c^{H\to L}$). For all samples and $B$ values there is a remarkable quantitative agreement between theory and experiment. We emphasize that the measured value of $J_c$ was not used in the heat-balance analysis and so our accurate prediction of $J_c$ is a good test for the validity of this theoretical framework. Note that this result shows that although the transition can theoretically occur anywhere between $I_c^{min}$ to $I_c^{max}$, in practice it occurs at $I_c^{min}$. In the discussion section and in Supplementary Note 8 we show that this is consistent with a switching wave that propagates through our superconductors[22].

## Discussion

Similarities with insulators - The heat-balance approach we used throughout this article is a general concept that can account for thermal bi-stabilities in various systems such as super-conductors[12,22,26–29], insulators[20,25], and even in earth's $T$[30,31]. Here its use was inspired by earlier studies of the $B$-driven insulating phase of a:InO[32]. There, the discontinuities in the $I-V$'s were attributed to bi-stable $T_{el}$ assuming that $\tilde{R}_{el-ph}$ dominates the electrons cooling rate at low $T$'s[20,25,33]. In Fig. 4a and b we plot $V$ vs. $I$ of one of our superconducting samples alongside $I$ vs. $V$ obtained from the $B$-driven insulating phase of a more disordered a:InO sample. The color-coding indicates the measurement $T$. We draw attention to the qualitative similarity between both measurements, and to the fact that the values of the parameters $\beta$ and $\Gamma$ do not vary significantly between superconducting and insulating samples (see Supplementary Table 1).

The Ohmic assumption: In our analysis, we assumed that all deviations from Ohmic transport are due to heating and other mechanisms leading to non-linearity, while present, are less effective and do not influence our main results. For example, we do not take into account intrinsic non-linearities that are known to exist in type-II superconductors at finite $T$ and $B$[5,34–36]. Our analysis, therefore, fails to quantitatively account for the onset of non-linearity at $I < I_c$ limiting its range of applicability to $I \geq I_c$ and $I \ll I_c$. In Supplementary Fig. 5, we demonstrate these deviations. A similar discrepancy was also reported in the heat-balance description of the insulating phase in a:InO[20,25]. We note that self-heating also applies in the presence of intrinsically non-linear effects and a complete description of our $I-V$'s awaits a theory that integrates both self-heating and intrinsic non-linearities.

Other mechanisms for $I_c$: The main result of this work is that, at low $T$'s, $I_c$ is a result of thermal bi-stability. While we clearly demonstrated this by accurately predicting $I_c$ under various conditions, it is also important to show that the other mechanisms for $I_c$ are not relevant in our experiments. We focus here on the role played by vortices at a finite $B$[4,6,7] and refer the reader to Supplementary Note 2 where we show that the mechanism of de-pairing of Cooper-pairs is unlikely.

To examine whether vortex de-pinning can be the mechanism causing our $I_c$'s we oriented $B$ in the sample's plane ($B_\parallel$) and conducted two measurements of $I_c$: one where $B_\parallel$ is aligned

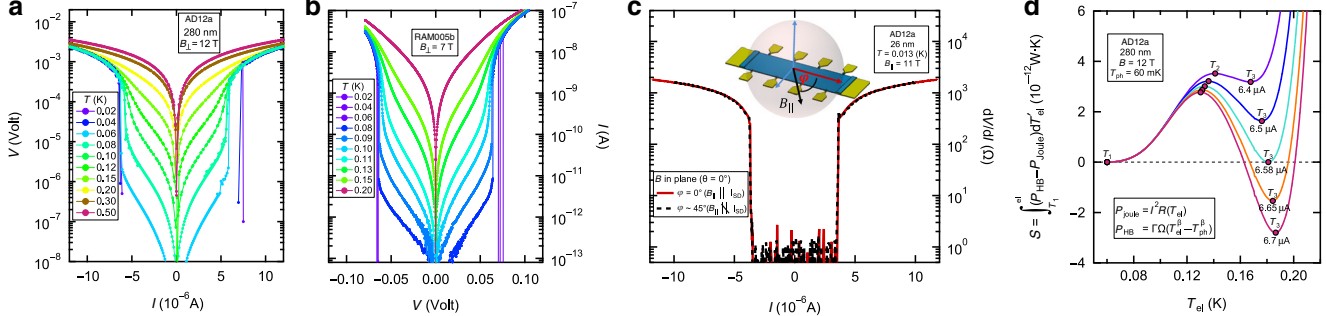

**Fig. 4 Critical currents. a** $V$ vs. $I$ obtained from the 280nm thick superconducting sample AD12a at $B = 12$ T. **b** $I$ vs. $V$ obtained from an insulating sample RAM005b at $B = 7$ T. **c** $dV/dI$ (log-scale) vs. $I$ at $T = 13$ mK and $B_{||} = 11$ T for the 26 nm thick sample at two in-plane angles $\varphi$ (defined in the cartoon) between $B_{||}$ and the source-drain current where the dashed black curve marks $\varphi \sim 45°$ and the red curve marks $\varphi = 0°$. **d** Extracting the minimum propagating current, $I_{\mathrm{p}}$, from the equal area condition (see text).

parallel to the source-drain $I$ ($I_{\mathrm{SD}}$) and one where $B_{||}$ was at an angle of $\varphi \approx 45°$ from $I_{\mathrm{SD}}$ ($\varphi$ is defined in the inset of Fig. 4c). Because the coherence length of our films $\xi \sim 5$ nm[37] is smaller than the film thickness vortices penetrating the plane of the sample experience a Lorentz force $\propto I_{\mathrm{SD}} \sin(\varphi)$. In Fig. 4c we plot $dV/dI$ vs. $I$ of the 26 nm thick sample at $T = 13$ mK and at $B_{||} = 11$ T where the dashed black line and the continuous red line correspond to $\varphi \approx 45°$ and $\varphi \approx 0°$, respectively. It is apparent that the entire $dV/dI$ curves, and in particular $I_{\mathrm{c}}$, are completely independent of $\varphi$ demonstrating that $I_{\mathrm{c}}$ is not due to collective de-pinning of vortices (similar insensitivity of transport properties to $\varphi$ was reported in high $T_{\mathrm{c}}$ superconductors[38–41]. While these results were still interpreted in terms of vortex motion, the different theoretical models rely heavily on the large anisotropy in high $T_{\mathrm{c}}$'s. The contrast between the $\varphi$ dependence of high $T_{\mathrm{c}}$'s and an amorphous MoGe alloy, which is a conventional type II superconductor, is demonstrated in ref. [41]).

Our $I_{\mathrm{c}}$ results are not different from those recently presented in[9]. These authors offered an interpretation very different from ours. They claim that $I_{\mathrm{c}}$ is a result of a combination of de-pairing and de-pinning. Their main experimental evidence are that $J_{\mathrm{c}} \propto |B - B_{\mathrm{c2}}|^{\alpha}$ with $\alpha \sim 1.6$ which is similar to the mean-field de-pairing value of 3/2 and that $J_{\mathrm{c}}$ is comparable to the de-pairing $J_{\mathrm{c}}$ (smaller by a factor of 4 according to their calculation). We do not intend to counter their claims. We do think, on the other hand, that our analysis better describes the data for three reasons: (1.) contrary to their results, the value of $\alpha$ is actually non-universal (see Supplementary Fig. 3). (2.) the de-pairing $J_{\mathrm{c}}$ is actually 10–15 times larger than their measured $J_{\mathrm{c}}$ and 10–400 times greater than in our measurements (see Supplementary Note 2). (3.) unlike their model the heat-balance analysis provides a good quantitative prediction to $J_{\mathrm{c}}$. In the supplementary material of ref. [9] Sacépé et al. discuss the possibility of a thermal bi-stability and provide several arguments against this interpretation. In Supplementary Note 7, we respond to these arguments.

Lack of hysteresis due to a propagating switching current: The heat-balance analysis can only determine the bounds of the $I$-interval, $I_{\mathrm{c}}^{\min}$ and $I_{\mathrm{c}}^{\max}$, where Equation (1) has two stable solutions[20]. The actual transition can occur anywhere within the interval $I_{\mathrm{c}}^{\min} \leq I_{\mathrm{c}}^{\mathrm{H} \rightarrow \mathrm{L}} \leq I_{\mathrm{c}}^{\mathrm{L} \rightarrow \mathrm{H}} \leq I_{\mathrm{c}}^{\max}$, depending on a dynamic interplay between the electrons, the phonons and the disorder where $I_{\mathrm{c}}^{\max}$ can be orders of magnitude greater than $I_{\mathrm{c}}^{\min}$ (see Supplementary Table 3). However, the limited hysteresis and the results displayed in Fig. 3c indicate that both $I_{\mathrm{c}}^{\mathrm{H} \rightarrow \mathrm{L}}$ and $I_{\mathrm{c}}^{\mathrm{L} \rightarrow \mathrm{H}}$ occur near $I_{\mathrm{c}}^{\min}$.

A common mechanism driving the transition between LR and HR states in bi-stable conductors (such as our samples at

$I_{\mathrm{c}}^{\min} < I_{\mathrm{c}}^{\max}$) is a propagating switching wave[22]. It turns out that such a switching wave is consistent with our observation that $I_{\mathrm{c}}^{\mathrm{H} \rightarrow \mathrm{L}} \approx I_{\mathrm{c}}^{\mathrm{L} \rightarrow \mathrm{H}} \approx I_{\mathrm{c}}^{\min}$. In this scenario, we consider a bi-stable material in the LR state (similar arguments apply to the HR state) where, due to disorder, there is local nucleation of high $T_{\mathrm{el}}$ domains embedded in the low $T_{\mathrm{el}}$ medium. In ref. [22] it is shown that above some minimum propagating current ($I_{\mathrm{p}}$) these hot $T_{\mathrm{el}}$ domains expand and that for $I < I_{\mathrm{p}}$ these domains shrink. $I_{\mathrm{p}}$ is extracted using the equal area condition $S(T_3, I) \equiv \int_{T_1}^{T_3}(P_{\mathrm{HB}} - P_{\mathrm{Joule}})dT_{\mathrm{el}} = 0$ where $P_{\mathrm{HB}} \equiv \Gamma\Omega(T_{\mathrm{el}}^{\beta} - T_{\mathrm{ph}}^{\beta})$, $P_{\mathrm{Joule}} = I^2R(T_{\mathrm{el}})$ and $T_1$ and $T_3$ are the low and high $T_{\mathrm{el}}$ solutions of Equation (1) respectively. In Fig. 4d we plot $S(T_{\mathrm{el}}, I)$ for several $I$ values and it can be seen that $I_{\mathrm{p}} = 6.58$ µA satisfies $S(T_3, I_{\mathrm{p}}) = 0$. Note hat $I_{\mathrm{p}} \sim I_{\mathrm{c}}^{\min} \ll I_{\mathrm{c}}^{\max}$, a result that accounts for our observation that $I_{\mathrm{c}}^{\mathrm{H} \rightarrow \mathrm{L}} \approx I_{\mathrm{c}}^{\mathrm{L} \rightarrow \mathrm{H}} \approx I_{\mathrm{c}}^{\min}$. In Supplementary Note 8, we provide a detailed analysis of how we extract $I_{\mathrm{p}}$.

Effects of the contacts: In the heat-balance analysis presented above we assumed that heating is a result of the non-vanishing $R$ of our type-II superconductor at finite $B$ and $T$. A different possibles source of heating that can potentially lead to the destruction of superconductivity at high $I$ is dissipation that originates at the contacts due to their finite $R$. To reduce contact $R$ we prepared our samples with a large overlap area of $333\mu m$ on $50\,\mu m$ between the source and drain Ti/au contacts and the a:InO (see "Methods"). We find this possibility unlikely for two reasons: First, in our heat-balance analysis we obtain $I_{\mathrm{c}}$ relying strictly on very low bias 4-terminal resistance measurements that are independent of contact resistances and were measured at $|I| \ll I_{\mathrm{c}}$ where heating is irrelevant. The accuracy of our analysis, as displayed in Fig. 3c, and its correspondence with the 4-terminal $R$ makes it unlikely that the effect we present is caused by heating at the contacts. We emphasize that because the electron thermal length in our samples at 10mK is $L_T \approx 0.2$ µm (approximated using a free electron gas approximation and a typical charge density of $n = 5 \times 10^{20}$ cm$^{-3}$), while our samples are 1mm long, the onset of the finite 4-terminal $R$ cannot be due to electron heating from the contacts. Second, because the contact $R$ is not typically strongly $B$ dependent, one would equally not expect $I_{\mathrm{c}}$ to be strongly $B$ dependent, which it clearly is, see Fig. 1c. Due to these arguments we find it more likely that the heating we report above is in the bulk of our finite $R$ type-II superconductor.

In summary, we have showed that the $I_{\mathrm{c}}$'s of superconducting a:InO films measured at low $T$'s and high $B$'s are well described by thermal bi-stabilities originating from a model of heat-balance between electrons and phonons (Equation (1)). Using this model we predicted quantitatively $I_{\mathrm{c}}$ for samples of different thicknesses for both $B_{\perp}$ and $B_{||}$.

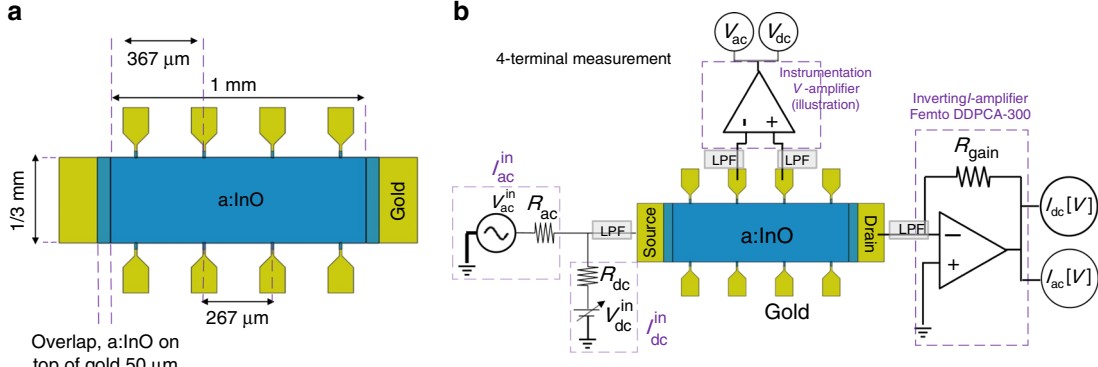

**Fig. 5 Sample geometry and schematics of transport measurements. a** Schematics of the sample geometry. **b** Schematics of the 4-terminal measurement scheme. A probing $I$ is driven between the sample's source and drain and the resulting $V$ drop is measured. An ac $I$ is produced by dropping an oscillating $V$ (from an EG&G 7265 lock-in amplifier) on a large resistor $R_{ac}$. To produce a dc $I$ we output a dc $V$ (from a Yokogawa 7651 programmable DC source) and drop it over a large resistor $R_{dc}$. $I$ from the drain is amplified using a variable gain Femto DDPCA 300 inverting $I$-amplifier. The measured $V$ drop is amplified using a home made instrumentation amplifier. The outputs of both $I$ and $V$-amplifiers are measured using an EG&G 7265 lock-in amplifier (for ac) and an HP 34401A multimeter (for dc). All lines are filtered using external low pass filters (LPF).

## Methods

**Sample fabrication**. a:InO was deposited in an Oxygen rich environment of $3 \times 10^{-5}$ Torr by e-gun evaporation of high purity $In_2O_3$ pellets onto a Si/SiO$_2$ substrate (a boron doped silicon wafer with $\rho < 5$ m$\Omega \cdot$ cm with a 580 nm thick oxide layer). The sample thickness was measured in situ during evaporation using a crystal monitor and verified later by atomic force microscopy. The contacts of the samples are Ti/Au, prepared via optical lithography prior to the $In_2O_3$ evaporation.

In Fig. 5a we present an illustration of the measured samples. The samples are Hall-bar shaped where the distance between source and drain contacts is 1 mm and the width of each sample is 1/3 mm. Adjacent $V$ contacts are located 0.8 squares apart (267 μm). Our study was performed using four such a:InO films of thicknesses 26, 57, 100, and 280 nm. Each sample was thermally annealed post deposition to a room-$T$ resistivity ($\rho$) of 4 $\pm$ 0.2 m$\Omega \cdot$ cm, which places them in the relatively low-disorder range of a:InO.

**Measurement setup**. The samples were measured in an Oxford instruments Kelvinox dilution refrigerator with a base $T$ of 10 mK, equipped with a z-axis magnet. In order to apply $B$'s in both perpendicular and in-plane orientations we mounted our samples on a probe with a rotating head. While measuring, all lines were filtered using room-$T$ RC filters with a cutoff frequency of 200 KHz.

The transport method we use to measure the zero bias $R$ (defined as $\lim_{I \to 0} \frac{V}{I}$) is a 4-terminal method. In this measurement, the sample is probed using an input ac $I$ (low frequency of ~ 10Hz) and we measure the resulting ac $V$ drop between a pair of contacts along the $I$ path. The 4-terminal configuration is described in Fig. 5b where during a zero bias $R$ measurement we fix $I_{dc}^{in} = 0$ and set a different $I_{ac}$ for samples of different thicknesses while maintaining a current density of $J \approx 0.1$ A $\cdot$ cm$^{-2}$. To measure a non-Ohmic response we use the same configuration as in Fig. 5b and measure how the differential resistance $\frac{dV}{dI} \equiv \frac{dV_{ac}}{dI_{ac}}$ varies as a function of $I \equiv I_{dc}^{in}$.

## Data availability

The data that support the findings of this study are available in Mendeley with the identifier https://doi.org/10.17632/24kvcdtkjn.1. The source data underlying Figs. 1b, c, 2a–c, 3a–c, 4a–d, and Supplementary Figs. 1a–e, 2, 3, 4a–g, 5a, b, 6b–f, 7a, b, 8a–e, and 9 a, b are provided as a Source data file.

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

## Acknowledgements

We are grateful to K. Michaeli, M.V Feigel'man, and B. Sacépé for fruitful discussions. This research was supported by The Israel Science Foundation (ISF Grant no. 556/17), the United States - Israel Binational Science Foundation (BSF Grant no. 2012210) and the Leona M. and Harry B. Helmsley Charitable Trust.

## Author contributions

A.D. and T.L. prepared the samples. A.D., T.L., I.T., and F.G. performed the experiments. A.D. and D.S. carried out the analysis, interpretation of the results, and wrote the paper with the input of all co-authors. D.S. supervised the project.

## Competing interests

The authors declare no competing interests.
