## [Peer Review File · Nature Communications]

Reviewers' comments:

Reviewer #1 (Remarks to the Author):

This work reports current-induced resistive switching in a:InO thin films. The authors observed some 4 orders of magnitude jumps in resistivity at a critical current I_c which was attributed to electron overheating. I find the qualitative arguments of the authors persuasive and the method of extracting the electron temperature new and interesting. The jumps shown in Fig. 1 are hardly consistent with the conventional vortex depinning, but rather indicative of switching between weakly and highly dissipative resistive state which can result from an electron overheating bistability. In my opinion the results are novel and can be of interest to a broader cond.mat. community. I can recommend publication of this work in Nature Communications after the following points have been addressed.

1. The critical current I_c was not clearly defined: it was mentioned in a passing comment that I_c is a "trapping" current but with no explanation of what it is. Fig 3 shows I_c as function of B but it was not explained in the text how I_c was actually obtained. The reader has to go all the way to the end of SM to find this information, which in my opinion should be moved to the main text as one of the central points. Given that Nature Comm. is a journal for a general physics audience, I think that the overheating bistability and the definition of I_c should be addressed in detail by moving Fig S4 from SI to the main text and showing it before the discussion of I_c results in Fig 3.

2. The lack of hysteresis seems inconsistent with the definition of I_c as a current at which a highly resistive state first appears. Hysteresis can occur at $I_c < I < I_2$ where I_2 is the current at which the low resistance state disappears. The authors should avoid the confusing jargon of "trapping" and "escaping" currents but clearly explain instead the meaning of I_c and I_2 using Fig. S4 in the main text. The scenario of "stochastic switching" between I_c and I_2 mentioned in sec D of the Discussion does not really explain the lack of hysteresis and why the resistive transition should occur at I_c but not at higher currents. The authors mentioned that hysteresis is weak so the difference between I_c and I_2 is not that significant, but Fig S4 suggests that $I_c = 6.2 \mu\text{A}$ but I_2 is certainly larger than $17 \mu\text{A}$ so the difference between I_c and I_2 is significant. The big resistive jumps and the lack of hysteresis may be more consistent with a switching wave between the cold and hot states propagating at $I = I_p$. In this case it is I_p which plays the role of the observed critical current. Here $I_c < I_p < I_2$, where I_p is defined by the "equal area theorem" in the heat balance equation (see Ref. 12 for details). For the case shown in Fig S4, I_p would be somewhat smaller than $17 \mu\text{A}$. Such switching wave could be initiated by either materials inhomogeneities or current leads.

3. The authors emphasize the importance of I_c for applications but a:InO films appear rather different from practical superconductors used in high current applications. I could not help but notice the anomalously large London penetration depth of 7 microns (35 times larger than in cuprates) resulting in a very small depairing current density $J_d = 40 \text{ kA/cm}^2$ (compared to typical J_d of the order of 100 MA/cm^2 in Nb and many other superconductors at $T \ll T_c$), and a huge GL parameter $k = 1400$ (more than an order of magnitude larger than in cuprates), based on the value of the kinetic inductance $L = 2 \text{ nH}$ mentioned in SI. If these numbers are correct, they may suggest that in more conventional materials J_c due to the electron overheating could be several orders of magnitude smaller than J_d at $T \ll T_c$.

4. The field dependence of the depairing current density near H_{c2} was calculated long ago by Boyd, Phys Rev. 145, 255 (1966). This work should be cited.

5. Are those a:InO films clean or dirty, that is, whether the m.f.p larger or smaller than ξ ? What is T_c and how broad the zero-field resistive transition is. A bit more info about the superconducting parameters would be useful.

6. The wavelength of thermal phonons at 20 mK is of the order of a few μm is much larger than the thicknesses of the a:InO films, so both the Kapitza conductance and the parameters β and γ should strongly depend on the film thickness. Yet the analysis in SI gives β varying non systematically from 5 to 10 (see table 1) with no clear trend of how β and γ evolve with the

film thickness. A discussion of this issue would be appropriate.

Reviewer #2 (Remarks to the Author):

The authors report low temperature critical current measurements in high resistance a:InO films. The main result of the experiments is that the critical current behavior and actual critical current values can be determined via a heat-balance model that has previously been used to explain breakdown in high resistance non-superconducting films. Overall, I find the authors analysis somewhat vague and essentially identical to what they have previously published. Moreover, in my view, there is a serious inconsistency in their interpretation of the measured critical current traces. I understand that once a sufficient amount of power is delivered to the electron gas, the gas temperature can suddenly jump to a very high value in a very high resistance non-superconducting system. This bistable behavior is predicted by Equation 1. But the authors fail to explain how power is delivered to the superconducting electron gas in the first place since $P=IV$ and V must be zero in the zero resistance state. Indeed, it is more reasonable to assume that once the superconducting state is compromised via pair-breaking, or vortex motion, or more likely heating at the contacts, then power begins to be delivered to the electron gas and the system quickly "runs away" to the normal state. So, in my view the heat-balance mechanism is not the source of the critical current behavior but instead a consequence of the breakdown of the zero resistance phase via some other mechanism. Some other comments are:

1. What role does the applied field play? Is the critical current mechanism the same in zero field for this heat-balance model?
2. More details need to be provided on how the measurements were done. If the critical currents were measured via dc currents then contact heating is almost certainly a concern. One usually uses a pulsed technique to circumvent Joule heating at the contacts.
3. The critical field behavior at low currents needs to be shown.

I do not recommend publication.

Reviewer #3:

In the manuscript, the authors presented the results of experimental studies of the critical current (I_c) of the a:InO superconducting film at low temperatures in magnetic fields close to the upper critical field. The current I_c was defined as the current at which the current-voltage characteristic has a voltage jump. When interpreting the experiments, the authors show that the value of I_c can be obtained from the heat balance equation (1). They believe that the voltage jump is of a thermal nature and occurs if the balance condition between heat generation and heat removal is violated.

Both the experiment results of the authors and the interpretation of the experiment results are of interest. It is especially important that the heat balance equation describes the experiment if the exponent $\beta = 5.1$, which is very close to the theoretical value $\beta = 5$ (see Refs.14, 15). Therefore, in my opinion, the article by A. Doron et al. entitled "The critical current of disordered superconductors near $T=0$ " can be published in Nature Communications. At the same time, I have two comments, the discussion of which could clarify the results of the article.

1) Equation (1) describes not only two stable states of a current-carrying superconducting film, but also two current values at which a voltage jump occurs (with increasing and decreasing current). The authors report that they observe "limited hysteresis" of critical currents. I believe that it is important to say more specifically which part of the current-voltage characteristic is described in terms of electron self-heating and where the discrepancy between theory and experiment begins.

2) The authors suggest that limited hysteresis of critical currents can be caused by the occurrence of hot spots. Indeed, regions with weakened superconducting properties may exist in inhomogeneous superconductors. When sufficiently large currents flow in these areas, the formation of resistive domains is possible. The theory of resistive domains is well known (see Ref. 12 and the references therein). It seems to me that it would be interesting to compare the results of the authors with the theory of resistive domains.

Response to referees

We thank the Reviewers for their thoughtful review and insightful comments regarding our manuscript. We would like to point out that both reviewers 1 and 3 support the publication of our findings. Reviewer 1 states *“I find the qualitative arguments of the authors persuasive and the method of extracting the electron temperature new and interesting. The jumps shown in Fig. 1 are hardly consistent with the conventional vortex depinning, but rather indicative of switching between weakly and highly dissipative resistive state which can result from an electron overheating bistability. In my opinion the results are novel and can be of interest to a broader cond.mat. community”* and reviewer 3 wrote *“Both the experiment results of the authors and the interpretation of the experiment results are of interest”*. Both reviewers recommended the publication of our manuscript after we address several points that they raised. We have addressed all of the points raised by the reviewers. One suggestion both reviewers 1 and 3 raised, regarding treating the lack of hysteresis using the theory of resistive domains, turned out to be consistent with our results, accounting for the lack of hysteresis. We thank both reviewers for this useful suggestion that improved our manuscript.

On the other hand, Reviewer 2 was not convinced by our results and raised several issues mostly regarding the source of dissipation in our system. We have addressed all of the issues the reviewer raised and made several clarifications in the manuscript.

In view of what we regard as strong support to our work by both reviewers 1 and 3, and our detailed response to every point all three reviewers made (listed below) we request that our manuscript will be published with minimum delay.

Below we begin by presenting the most significant change made following the suggestion that was common to both reviewers 1 and 3, then we reply to each comment made by each reviewer and we end with a summary of the changes made in the new version of the manuscript.

Response to a common comment of reviewers 1 and 3:

Both reviewers 1 and 3 had a similar suggestion that appears to be very relevant. Reviewer 1 wrote: *“The big resistive jumps and the lack of hysteresis may be more consistent with a switching wave between the cold and hot states propagating at $I=I_p$. In this case it is I_p which plays the role of the observed critical current. Here $I_c < I_p < I_2$, where I_p is defined by the “equal area theorem” in the heat balance equation (see Ref. 12 for details). For the case shown in Fig S4, I_p would be somewhat smaller than $17 \mu\text{A}$. Such switching wave could be initiated by either materials inhomogeneities or current leads”*.

Reviewer 3 had a similar suggestion: *“The authors suggest that limited hysteresis of critical currents can be caused by the occurrence of hot spots. Indeed, regions with weakened superconducting properties may exist in inhomogeneous superconductors. When sufficiently large currents flow in these areas, the formation of resistive domains is possible. The theory of resistive domains is well known (see Ref. 12 and the references therein). It seems to me that it would be interesting to compare the results of the authors with the theory of resistive domains.”*

This is a very interesting point that is consistent with our lack of hysteresis. We have considered this possibility initially and wrongfully chose to dismiss it, because we naively expected I_p to be more or less the average between the lower and upper theoretical bounds of the bi-stability I interval (found from the graphical solution of the heat-balance equation), which in the new version of the manuscript we define as I_c^{min} and I_c^{max} . As can be seen in the table

below, $I_c^{max} \gg I_c^{min}$. According to the suggestion of the reviewers we followed Ref 12, calculated I_p , and found that actually I_p is similar to I_c^{min} .

T_{ph}	I_c^{max}	I_c^{min}	I_p
80mK	12 μA	6.2 μA	6.41 μA
70mK	20 μA	6.2 μA	6.51 μA
60mK	41 μA	6.2 μA	6.58 μA
50mK	140 μA	6.2 μA	6.61 μA
40mK	1.1mA	6.2 μA	6.63 μA

The central part of our analysis is detailed in Figure 2a-e (displayed in the next page) and we also changed section D of the discussion section in the main-text and added to the supplemental material a new section, Sec. S8, titled “Lack of Hysteresis - Propagating switching waves”.

In chapter 3 of Ref. 12 the authors discuss the propagation of a switching wave in bi-stable conductors. They write that “in most cases the transitions between these states (these phases)” [the cold and hot T_{el} phases] “are initiated by local perturbations, which result in the nucleation of a new phase, which then propagates to cover the whole specimen”. They proceed to assume that in the sample there are domains at a T of T_1 and of $T_3 > T_1$ and calculate the velocity of the domain wall as a function of I . They show that for $I > I_p$ the domain with T_3 will expand and that it will collapse at $I < I_p$ where I_p is found from the condition $\int_{T_1(I)}^{T_3(I)} (P_{HB} - P_{Joule}(I)) \kappa(T_{el}) dT'_{el} = 0$ where κ is the thermal conductivity, $P_{HB} = \Gamma \Omega (T_{el}^\beta - T_{ph}^\beta)$ and $P_{Joule} = I^2 R(T_{el})$. For simplicity we will neglect the T_{el} dependence of κ which simplifies the condition to $\int_{T_1}^{T_3} (P_{HB} - P_{Joule}) dT_{el} = 0$.

Figure 1: Extracting I_p from equal area law. All figures correspond to the 280nm thick sample at $B = 12\text{T}$ and $T_{ph} = 60\text{mK}$. **a.** $P_{HB} = \Gamma\Omega(T_{el}^\beta - T_{ph}^\beta)$ (red) and $P_{Joule} = I^2 R(T_{el})$ vs T_{el} plotted on a log-log scale for $I = 8\mu\text{A}$. We mark the two stable solutions as T_1 and T_3 and the unstable solution as T_2 . The equal area rule is satisfied if the two shaded areas in the figure are equal. Changing I will move the green curve as a whole without changing its shape therefore increasing I will decrease the shaded area between T_1 and T_2 and increase the area between T_2 and T_3 . From the figure it appears that the area between T_1 and T_2 is greater than the area between T_2 and T_3 , this is actually an artifact of plotting this figure on a log-log scale. **b.** Here we plot the same data as in **a** on a linear scale. It can be seen that the shaded area between T_2 and T_3 is actually much greater than the area between T_1 and T_2 therefore I_p that meets the equal area law should be smaller than $8\mu\text{A}$. Note that here $I_c^{min} \approx 6.2\mu\text{A}$ and $I_c^{max} \approx 41\mu\text{A}$ therefore we see that $I_p \sim I_c^{min}$ which can account for the lack of hysteresis. **c.** $P_{HB} - P_{Joule}$ from Figure **b**. **d.** $S(T_{el}, I) \equiv \int_{T_1}^{T_{el}} (P_{HB} - P_{Joule}) dT'_{el}$, which is the integrated area between the curves from T_1 up to some T_{el} vs. T_{el} . I_p that satisfies the equal area rule is found by the condition $S(T_3, I_p) \equiv \int_{T_1}^{T_3} (P_{HB} - P_{Joule}) dT'_{el} = 0$. **e.** $S(T_{el}, I)$ vs T_{el} plotted for different values of I between $6.4\mu\text{A}$ and $6.7\mu\text{A}$. It can be seen that $S(T_3, 6.58\mu\text{A}) = 0$ therefore $I_p = 6.58\mu\text{A}$.

Response to the comments of Reviewer 1:

1. The reviewer writes: *“The authors should avoid the confusing jargon of “trapping” and “escaping” currents but clearly explain instead the meaning of I_c and I_2 using Fig. S4 in the main text.”*

Answer: We have made the appropriate changes and defined the I_c 's according the states direction of the transition between the low resistive (LR) and high resistive (HR) states. We define I_c of the transition from the LR to the HR state as $I_c^{L \rightarrow H}$ and the transition from the HR to the LR state as $I_c^{H \rightarrow L}$.

2. The reviewer writes: *“The critical current I_c was not clearly defined: it was mentioned in a passing comment that I_c is a “trapping” current but with no explanation of what it is. Fig 3 shows I_c as function of B but it was not explained in the text how I_c was actually obtained. The reader has to go all the way to the end of SM to find this information, which in my opinion should be moved to the main text as one of the central points. Given that Nature Comm. is a journal for a general physics audience, I think that the overheating bistability and the definition of I_c should be addressed in detail by moving Fig S4 from SI to the main text and showing it before the discussion of I_c results in Fig 3.”*

Answer: We agree with the reviewer's comment. In the revised version we added a subfigure to Fig 1 (Fig 1b) where we plot the same data that was displayed in Fig S4 of the previous SM (which is Fig S7 in the revised version) on both linear and logarithmic scales. In this subfigure we mark the low resistive (LR) and high resistive (HR) states, and we define the transitions from the HR to the LR state as $I_c^{H \rightarrow L}$ (in the previous version we referred to this transition as the “trapping” current) and from the LR to the HR state as $I_c^{L \rightarrow H}$ (previously referred to as “escaping” current). In the main text describing the figure we clarified that because in our measurements the hysteresis is not pronounced and $I_c^{H \rightarrow L} \approx I_c^{L \rightarrow H}$ (we further discuss the hysteresis in your next comment), throughout the manuscript, wherever there is no need to distinguish between $I_c^{H \rightarrow L}$ and $I_c^{L \rightarrow H}$ we refer to both of them as I_c . In addition, in the text describing Fig 3 we added an explanation that the experimental I_c used in the figure is $I_c^{H \rightarrow L}$ and that the plotted theoretical I_c is I_c^{min} , which is the lowest I where the heat-balance equation has two stable solutions.

3. The reviewer writes: *“The lack of hysteresis seems inconsistent with the definition of I_c as a current at which a highly resistive state first appears. Hysteresis can occur at $I_c < I < I_2$ where I_2 is the current at which the low resistance state disappears.” ... “The scenario of “stochastic switching” between I_c and I_2 mentioned in sec D of the Discussion ... does not really explain the lack of hysteresis and why the resistive transition should occur at I_c but not at higher currents. “*

Answer: We agree that the theoretical approach we follow does not prohibit hysteresis. The exact values of the measured critical currents are not predicted by the heat-balance model, from which we can only extract an interval of currents, $I \in [I_c^{min}, I_c^{max}]$, where the heat balance equation has two stable solutions (if $I < I_c^{min}$ the system can only be in the low-resistive state and if $I > I_c^{max}$ it can only be in the high-resistive state). The model sets the following inequality: $I_c^{min} \leq I_c^{H \rightarrow L} \leq I_c^{L \rightarrow H} \leq I_c^{max}$. What we measure in practice is that $I_c^{min} \approx I_c^{H \rightarrow L} \approx I_c^{L \rightarrow H}$ which means that not only there is almost no hysteresis but both transitions occur “prematurely”, namely near the low end of the bi-stable current interval $[I_c^{min}, I_c^{max}]$. In the added analysis suggested by the reviewer (see above) we see that an I_p that satisfies a non-equilibrium variant of the equal area law is also very similar to I_c^{min} .

In the previous submission we wrote that the transition can occur stochastically anywhere within this interval, by “stochastically” we meant that a priori the measured I_c is not determined by the theory, only the boundaries on I_c are determined. We believe that our terminology might have been misleading and we have avoided using the term stochastic in the new version. We also elaborated more on the relation between the theoretically predicted I_c^{min}, I_c^{max} and the measured $I_c^{H \rightarrow L}, I_c^{L \rightarrow H}$ in order to help the reader understand.

4. The reviewer writes: “The authors mentioned that hysteresis is weak so the difference between I_c and I_2 is not that significant, but Fig S4 suggests that $I_c = 6.2 \mu A$ but I_2 is certainly larger than $17 \mu A$ so the difference between I_c and I_2 is significant.”

Answer: We interpret the reviewer’s definition of I_2 as $I_c^{L \rightarrow H}$, which is a measured quantity. If this is indeed what the reviewer meant, we think that the statement is inaccurate and we would like to explain Fig S4b of the previous version again (note that in the revised version it is Fig S7b of the SM where there is an additional $I^2 R(T_{el})$ curve corresponding to $I = I_c^{max}$, for convenience we plot below the new version of the figure). Our goal in displaying this figure is to solve the heat-balance equation graphically and extract T_{el} as a function of I and T_{ph} . The red curve marks the right hand side of the heat balance equation $P_{HB} \equiv \Gamma \Omega (T_{el}^\beta - T_{ph}^\beta)$ where we use the values of $\Gamma \Omega$ and β extracted from the prior heat-balance analysis (summarized in Fig 2 of the main-text and detailed in Sec S4 of the new version of the SM), $T_{ph} = 60 \text{ mK}$ (T_{ph} of the measured $\frac{dV}{dI}$ vs I in Fig S7a) is kept constant and T_{el} is the parameter we want to solve for. This red curve does not include any measured data. The four other curves (purple, blue, green and black) mark the left hand side of the heat balance equation, $P_{Joule} \equiv I \cdot V = I^2 R(T_{el})$, at four different I values ($I = -3 \mu A, -6.2 \mu A, -17 \mu A, -41 \mu A$) where here $R(T_{el})$ is the measured $R(T)$ at zero-bias ($R(I = 0.1 \mu A \ll I_c \sim 6.2 \mu A)$) and I is a parameter we vary while solving for T_{el} . A solution of the heat balance equation at a given I is where the P_{Joule} curve corresponding to I intersects with the red P_{HB} curve (i.e. the two sides of the equations are equal), we track these crossing points while varying the parameter I .

Starting from the purple curve which corresponds to $I = -3 \mu A$ we see that the red and purple curves intersect once, at $T_{el} \sim T_{ph}$ i.e. $T_{el}(I = -3 \mu A, T_{ph} = 60 \text{ mK}) \sim 60 \text{ mK}$. As the figure is plotted on a log scale, while increasing $|I|$ further, $I^2 R$ maintains the same shape and only moves vertically as a whole. At $I = -6.2 \mu A$ (blue line) the two sides of the equation intersect twice, once as before at $T_{el} \sim T_{ph}$ and once at $T_{el} \sim 150 \text{ mK}$ where the two curves are tangent. This value of I is the lowest I where there are two stable solutions for the heat-balance equation, i.e. $I_c^{min} = 6.2 \mu A$. In practice we see (for example in light blue in Fig S7a) that this theoretically predicted I_c^{min} is very similar to the measured quantities $I_c^{H \rightarrow L}$ and $I_c^{L \rightarrow H}$ (which we now understand as $I_p \approx I_c^{max}$). We emphasize again, $I_c^{H \rightarrow L} \approx I_c^{L \rightarrow H} \approx I_c^{H \rightarrow L}$ is not a priori set by the theory, the fact that it holds suggests that for some reason (maybe due to the equal area rule) our system “prefers” to be in the HR state whenever such a state exists. The green curve marks P_{Joule} at $I = -17 \mu A$. It can be seen that there are three crossing points between the green and red curves corresponding to three solutions to the heat-balance

equation (the middle solution is unstable). As there are three solutions it means that $I = -17 \mu A$ is still in the bi-stability I interval $I_c^{min} \leq |I| \leq I_c^{max}$ therefore theoretically the system could still “choose” to be in the LR state with $T_{el} \sim T_{ph}$ or at the HR state with $T_{el}(-17 \mu A) \sim 350 mK$. We emphasize that the crossing point at $T_{el}(-17 \mu A) \sim 350 mK$ does not mean that I_2 , which we interpret as the measured $I_c^{L \rightarrow H}$, is $17 \mu A$, it only means that $17 \mu A < I_c^{max}$. In practice $I_2 \equiv I_c^{L \rightarrow H}$ is $\sim 6.2 \mu A$. The black curve marks P_{joule} at $I = I_c^{max} = -41 \mu A$. It can be seen that there are only two crossing points between the black and red curves where for $I > I_c^{max}$ the low T_{el} solution does not exist.

Figure 2: Displayed here is another version of Fig S7b (S4b in the previous version) where in addition to red curve corresponding to $\Gamma\Omega(T_{el}^\beta - T_{ph}^\beta)$ and the purple, blue and green curves corresponding to $I^2 R(T_{el})$ at $I = -3, -6.2, -17 \mu A$ we added a fourth black $I^2 R(T_{el})$ curve corresponding to $I = -41 \mu A$ which is the upper bound of the bi-stability interval I_c^{max} . It can be seen that at this I value the low black and red curves are tangent at the low T_{el} solution and if we increase $|I|$ beyond it the low T_{el} solution will not exist anymore leaving only the high T_{el} solution.

- The reviewer writes: “The big resistive jumps and the lack of hysteresis may be more consistent with a switching wave between the cold and hot states propagating at $l=lp$. In this case it is lp which plays the role of the observed critical current. Here $I_c < lp < I_2$, where lp is defined by the “equal area theorem” in the heat balance equation (see Ref. 12 for details). For the case shown in Fig S4, lp would be somewhat smaller than $17 \mu A$. Such switching wave could be initiated by either materials inhomogeneities or current leads.”

Answer: See “Response to common comment of reviewer 1 and 3” above.

- The reviewer writes: “The authors emphasize the importance of I_c for applications but a:InO films appear rather different from practical superconductors used in high current applications. I could not help but notice the anomalously large London penetration depth of 7 microns (35 times larger than

in cuprates) resulting in a very small depairing current density $J_d = 40 \text{ kA/cm}^2$ (compared to typical J_d of the order of 100 MA/cm^2 in Nb and many other superconductors at $T \ll T_c$), and a huge GL parameter $k = 1400$ (more than an order of magnitude larger than in cuprates), based on the value of the kinetic inductance $L = 2 \text{ nH}$ mentioned in SI. If these numbers are correct, they may suggest that in more conventional materials J_c due to the electron overheating could be several orders of magnitude smaller than J_d at $T \ll T_c$ ”

Answer: We agree with the reviewer and in fact the possibility that J_c in high T_c superconductors due to electron heating might be much smaller than J_d has been previously studied. One important manifestation of heating effects in high T_c superconductors is “thermal runaway” which is crucial in application such as superconducting magnets [1] [2] where the substance is heated as a whole relative to the coolant. This problem is typically solved by cladding such materials with a metal like copper that has good thermal conductivity. Heating due to electron-phonon de-coupling was studied in YBCO [3] [4] [5] where Ref. [3] concludes that near T_c J_c is due to a flux-flow instability and at low T 's it is due to what they refer to as the “hot-spots effect” which is Joule-heating (the term is taken from Ref. [6]).

7. The reviewer writes: “The field dependence of the depairing current density near H_{c2} was calculated long ago by Boyd, Phys Rev. 145, 255 (1966). This work should be cited”

Answer: Thank you for the correction, we now cite this work appropriately.

8. The reviewer writes: “Are those a:InO films clean or dirty, that is, whether the m.f.p larger or smaller than ξ ? What is T_c and how broad the zero-field resistive transition is. A bit more info about the superconducting parameters would be useful”

Answer: Relative to typical highly disordered a:InO films the films studied here are considered to be of low disorder level. Having said that they are still highly disordered superconductors with a m.f.p of less than 1nm while $\xi \sim 5\text{nm}$ (using the Drude formula: our samples have $\rho = 4\text{m}\Omega \cdot \text{cm}$ and assuming a charge density typical for our a:InO films of $n = 5 \cdot 10^{20} \text{ cm}^{-3}$ we get a mean free path of $\sim 0.5\text{nm}$ where ξ calculated from $H_{c2} \sim 12\text{T}$ is $\sim 5\text{nm}$). Unfortunately we have measured T_c only for one of the samples (the 26nm thick sample) where $T_c = 2.4\text{K}$ and the width of the transition (between 90% and 10% of the normal state resistance) is $\Delta T_c \approx 0.35\text{K}$. The reason we have measured only for one sample is technical, as we are measuring in a dilution refrigerator it takes a special effort to reach T 's of above 1.5K. We assume that T_c of the other films is similar. We have added all these values to Sec. S1 of the supplemental material.

9. The reviewer writes: “The wavelength of thermal phonons at 20 mK is of the order of a few μm is much larger than the thicknesses of the a:InO films, so both the Kapitza conductance and the parameters beta and gamma should strongly depend on the film thickness. Yet the analysis in SI gives beta varying non systematically from 5 to 10 (see table 1) with no clear trend of how beta and Gamma evolve with the film thickness. A discussion of this issue would be appropriate”

Answer: We would like to clarify that according to our analysis, presented in Sec. S6 of the current version of the SM, the thermal bottleneck is the electron-phonon coupling and not the Kapitza resistance. The way we show it is by actually measuring what the Kapitza resistance

is (which happened to fit a known functional form with not fit parameters) and then showing that it cannot account for the increase in T_{el} in our measurements. For the electron-phonon coupling the parameters β and Γ which are presented in table 1 are independent of the sample's thickness and only Ω which is the volume should increase with the thickness.

Response to the comments of Reviewer 2:

1. The reviewer writes: *“in my view, there is a serious inconsistency in their interpretation of the measured critical current traces. I understand that once a sufficient amount of power is delivered to the electron gas, the gas temperature can suddenly jump to a very high value in a very high resistance non-superconducting system. This bistable behavior is predicted by Equation 1. But the authors fail to explain how power is delivered to the superconducting electron gas in the first place since $P=IV$ and V must be zero in the zero resistance state. Indeed, it is more reasonable to assume that once the superconducting state is compromised via pair-breaking, or vortex motion, or more likely heating at the contacts, then power begins to be delivered to the electron gas and the system quickly “runs away” to the normal state. So, in my view the heat-balance mechanism is not the source of the critical current behavior but instead a consequence of the breakdown of the zero resistance phase via some other mechanism”*

Answer: This is indeed a good point and in the initial submission we did not explain how power is delivered to the superconducting electrons in the first place. Following this comment, we added a short explanation of this issue in the main-text. The point that was missing to the reviewer is that highly disordered, thin film type-II superconductors in the presence of a magnetic field above H_{c1} (H_{c1} is practically zero for a:InO) are expected to have zero resistance only at $T = 0$ [7] [8] therefore, as we are measuring at a finite T , there is no true zero resistance state. For example, In Fig. 1b of the new submission we plot $\frac{dV}{dI}$ vs. I_{dc} of our sample at $B = 12T$ and $T = 60mK$ where the main-frame is on a linear scale and it seems that the LR state is a zero resistance state while in the inset we plot the same data on a logarithmic scale showing that there is some small but finite residual resistance (of $\sim 0.1\Omega$) even at low I 's which are orders of magnitude below I_c . We emphasize that this measurements is a four-terminal measurement were the source and drain leads are 1/3 of a mm away from the voltage measuring leads therefore the measured residual resistance at low currents cannot be due to the contacts. This residual resistance transfers power to the superconducting electron gas.

2. The reviewer writes: *“What role does the applied field play? Is the critical current mechanism the same in zero field for this heat-balance model?”*

Answer: This is a good question, unfortunately we are not sure what role the field plays. At zero field we are unable to characterize I_c due to technical limitations, in order to measure a response we needed to use I 's that are of several mA's, the problem with that is that our refrigerator is wired with resistive constantan wires with a resistance of $\sim 50\Omega$ inside the mixing chamber therefore at $1mA$ the dissipation from the constantan wires in our mixing chamber is comparable to the cooling power of the refrigerator and the mixing chamber heats up as a whole.

3. The reviewer writes: *“More details need to be provided on how the measurements were done. If the critical currents were measured via dc currents then contact heating is almost certainly a concern. One usually uses a pulsed technique to circumvent Joule heating at the contacts”*

Answer: First, following the request for additional details on how the measurements were done we have added to the supplemental material an “Experimental setup” section (Sec. S2) where we describe our transport setup. Regarding the use of a pulsed technique, it is a good and interesting idea to compare our results, acquired by combining a constant dc- I with a low frequency ac- I (which is a standard method, for example see Refs. [9] [4] [10]) with results using a pulsed technique. At the time being our experimental setup is not suited for such measurements but we plan to make the appropriate adjustments. We would also like to note that pulsed methods have several shortcomings [11]. Regarding the heating occurring at the contacts, although we can never completely rule out the possibility that the nucleation of a hot T_{el} domain occurs at the contacts, we find it rather unlikely for three reasons:

- i. If one would expect the measured I_c 's to be purely a result of heating at the contacts, we would not expect I_c to have a B dependence. As plotted in Fig. 1c I_c is strongly B dependent. If the reviewer's claim is that the role of the contact resistance is to cause a local nucleation of a hot domain that eventually expands to the whole sample then although we cannot rule out this possibility, we think that this mechanism is not necessary here as our highly disordered superconductor has a small residual resistance in the LR state (as we wrote above) which already acts as a source for dissipation.
- ii. The source and drain contacts of our sample are rather large and they have an overlap area of $333\mu\text{m}$ on $50\mu\text{m}$ with the a:InO which is evaporated on top of the contacts (see Fig. S1 of the new version of the supplemental material which is a sketch of the sample and its contacts). The measurements we perform are 4-terminal measurements where we measure the voltage drop between contacts that are at a distance of $367\mu\text{m}$ away from the source and drain contacts.
- iii. The fact that we show that we can use the 4-terminal resistance measurement, which does not measure the contact resistance and was measured at low I 's (much below I_c) where heating is not relevant, to predict I_c makes it very unlikely that the effect we present is caused by heating at the contacts.

4. The reviewer writes: *“The critical field behavior at low currents needs to be shown”*

Answer: Good idea, thank you, we have added a Fig. S3 where we plot the critical field behavior as a function of T at low currents for all samples.

Response to the comments of Reviewer 3:

1. The reviewer writes: *“Equation (1) describes not only two stable states of a current-carrying superconducting film, but also two current values at which a voltage jump occurs (with increasing and decreasing current). The authors report that they observe “limited hysteresis” of critical currents. I believe that it is important to say more specifically which part of the current-voltage characteristic is*

described in terms of electron self-heating and where the discrepancy between theory and experiment begins”

Answer: This is a very good point, in the supplemental material of our new submission we have added Fig. S8 where we compare between the measured and simulated results and focus on these deviations. We reference this figure from the discussion regarding the deviations due to the simplified Ohmic approximation in the main-text.

2. The reviewer writes: “The authors suggest that limited hysteresis of critical currents can be caused by the occurrence of hot spots. Indeed, regions with weakened superconducting properties may exist in inhomogeneous superconductors. When sufficiently large currents flow in these areas, the formation of resistive domains is possible. The theory of resistive domains is well known (see Ref. 12 and the references therein). It seems to me that it would be interesting to compare the results of the authors with the theory of resistive domains”

Answer: See “Response to common comment of reviewer 1 and 3” above.

Summary of the main changes in the manuscript:

- We changed our terminology from trapping and escape critical current to $I_c^{H \rightarrow L}$, $I_c^{L \rightarrow H}$ (following a comment of reviewer 1).
- We added Fig.1b where we define the $I_c^{H \rightarrow L}$, $I_c^{L \rightarrow H}$ (following a comment of reviewer 1).
- We defined I_c^{min} and I_c^{max} as the theoretical limits of the bi-stability interval (following a comment of reviewer 1).
- We have added (see discussion section D of main-text and supplemental material Sec. S8) an analysis showing that the observation that $I_c^{H \rightarrow L} \approx I_c^{L \rightarrow H} \approx I_c^{min}$ is consistent with a propagation of a switching wave in our bi-stable conductor (as suggested by reviewers 1 and 3).
- We have added to Fig.S7b the black curve which corresponds to I_c^{max} (following a comment of reviewer 1).
- We have corrected the citation to Eq.S1 (following a comment of reviewer 1).
- We have added to Sec. S1 of the supplemental material the values of the mean-free-path, T_c , width of T_c and the coherence length (following a comment of reviewer 1).
- We have emphasized that in highly disordered, thin film type-II superconductors in the presence of a magnetic field above H_{c1} (H_{c1} is practically zero for a:InO) are expected to have zero resistance only at $T = 0$. We also show that in the inset of Fig.1b (following a comment of reviewer 2).
- We added to the supplemental material an “Experimental setup” section (Sec. S2) where we describe our transport setup (following a comment of reviewer 2).
- We have added a Fig. S3 where we plot the critical field behavior as a function of T at low currents for all samples (following a comment of reviewer 2).
- We added Fig. S8 where we compare between the measured and simulated results and focus on the deviations at low I 's (following a comment of reviewer 3).

References

- [1] H. Maeda and Y. Yoshinori, "Recent developments in high-temperature superconducting magnet technology," *IEEE Transactions on Applied Superconductivity*, vol. 24, no. 3, pp. 1--12, 2013.
- [2] M. Tinkham, Introduction to superconductivity, Courier Corporation, 2004, pp. 186-187.
- [3] Z. Xiao, E. Andrei and P. Ziemann, "Coexistence of the hot-spot effect and flux-flow instability in high-T_c superconducting films," *Physical Review B*, vol. 58, no. 17, p. 11185, 1998.
- [4] J. Viña, M. González, M. Ruibal, S. Currás, J. Veira, J. Maza and F. Vidal, "Self-heating effects on the transition to a highly dissipative state at high current density in superconducting YBa₂Cu₃O_{7- δ} thin films," *Physical Review B*, vol. 68, no. 22, p. 224506, 2003.
- [5] M. N. Kunchur, "Unstable flux flow due to heated electrons in superconducting films," *Physical review letters*, vol. 89, no. 13, p. 137005, 2002.
- [6] W. Skocpol, M. Beasley and M. Tinkham, "Self-heating hotspots in superconducting thin-film microbridges," *Journal of Applied Physics*, vol. 45, no. 9, pp. 4054--4066, 1974.
- [7] G. Blatter, M. V. Feigel'man, V. B. Geshkenbein, A. I. Larkin and V. M. Vinokur, "Vortices in high-temperature superconductors," *Reviews of Modern Physics*, vol. 66, no. 4, p. 1125, 1994.
- [8] M. P. A. Fisher, "Quantum Phase Transitions in Disordered Two-Dimensional Superconductors," *Phys. Rev. Lett.*, vol. 65, p. 923, 1990.
- [9] B. Sacépé, J. Seidemann, F. Gay, K. Davenport, A. Rogachev, M. Oviaia, K. Michaeli and M. V. Feigel'man, "Low-temperature anomaly in disordered superconductors near B_{c2} as a vortex-glass property," *Nature physics*, vol. 15, p. 48, 2019.
- [10] Y. Cao, V. Fatemi, S. Fang, K. Watanabe, T. Taniguchi, E. Kaxiras and P. Jarillo-Herrero, "Unconventional superconductivity in magic-angle graphene superlattices," *Nature*, vol. 556, no. 7699, p. 43, 2018.
- [11] M. Woźniak, S. C. Hopkins, B. A. Głowacki and T. Janowski, "Comparison of DC and pulsed critical current characterisation of NbTi superconducting wires," *Prz. Elektrotech*, vol. 85, pp. 198--199, 2009.

Reviewers' comments:

Reviewer #1 (Remarks to the Author):

In the revised version some points of my previous report have been addressed, although mostly in the supplemental material (SM). The main text still has presentation issues and does not really convey the main message to a general readership of Nature Communications. It sounds like the authors just postulate the electron overheating scenario and try to proselytize the reader by making general statements and talking about different critical currents without clear explanation what they are. It is unfortunate because plenty of convincing experimental evidences are given in the SM which appears to be more of a real research paper than the main text.

Most figures with experimental data in the main text show redundant (and sometimes duplicate) semilog plots of dV/dI from which the overheating bistability does not readily follow. Then all of a sudden the authors show the key Fig. 3 and claim that the overheating model is in excellent agreement with experiment. I do not know how many people will be convinced by this way of presentation, but in my opinion the key experimental evidences of the electron overheating bistability should be presented in the main text before Fig. 3. One way of doing so would be to show an example of graphic solution of the thermal balance eq based on the analysis of experimental data (similar to Fig. S7b in the SM) and use it not only to clearly demonstrate the thermal bistability and the magnitude of electron overheating, but also to explain how the minimum and maximum currents I_c^{\min} and I_c^{\max} are defined, and what are the equal area theorem and the associated current I_p in one sentence, relegating all technical details of analysis of particular cases to the SM. This would also help the reader to understand the physics of switching waves and the lack of hysteresis mentioned in the discussion section without going back and forth between the main text and the SM on each principal issue. I think that this work can be of interest to the Nat Com readership, but the authors should make an effort to convince not just a few specialists but a broader condensed matter superconductivity/condmat community that the overheating scenario is indeed essential.

Some inaccurate/peculiar statements should be fixed: page 1: "Lorentz force acting on the vortices exceeds their binding energy" – this sounds like current somehow unbind pairs of vortices, not to mention that a force cannot exceed energy because they have different dimensionalities. Perhaps, something like "the Lorentz force acting on vortices exceeds their depinning force" would be more informative. Page 1. "near the high critical B" – why cannot the standard "near the upper critical field" be used? Also, the list of references was cut and pasted from the SI, but Refs. 44-59 were not mentioned in the main text.

Reviewer #2 (Remarks to the Author):

Dear Editor-

I have considered the authors' response to my criticisms of the original manuscript. Unfortunately, I am unpersuaded. Contact heating is a very serious concern in critical current measurements. This was well understood more than 50 years ago by one of the pioneers of experimental studies of thin film superconductors, Rolf Glover III, who stated "The second trouble is due to warming of the film by Joule heating which can occur if small normally conducting regions are present, for example where electrical contact is made to the film. The problem is acute since the current densities exceed 10^6 A/cm²" (Rev. Mod. Phys. 36, 299 (1964).) Indeed, I have personally measured critical currents in thin film superconductors and found that low duty cycle pulse measurements can give critical current

values that are more than an order of magnitude greater than those of a dc measurement. There are groups all over the world that have the capability to perform pulsed critical current measurements so not having the capability "in house" is not a legitimate excuse. The critical current data in the manuscript must be verified via pulsed measurements or, perhaps by incorporating superconducting contacts, before the authors rush to publication.

I do not recommend publication in Nature Communications nor in any other reputable journal for that matter.

Reviewer #3 (Remarks to the Author):

The authors gave acceptable answers to all my comments and significantly clarified the manuscript. After an additional analysis, the authors argue that it is the motion of the switching wave between the S and N states in the a:InO film that is consistent with their observations. In my opinion, the revised manuscript NCOMMS-19-30499A can be published in Nature Communications.

Response to referees

Summary of the main changes in the manuscript:

- Following the suggestion of Reviewer 1 the graphical solution section of the supplemental material (Fig.S7a and S7b and their description of the previous submission) was moved to the main text (Fig.3a and 3b). In addition to the figures the graphical solution is also described in the text at the end of the 3rd page and at the beginning of page 4.
- Following the suggestion of Reviewer 1 we added to the main text Fig.4d where we show how to extract the minimum propagation current I_p .
- Following the suggestion of Reviewer 1 and the comments of Reviewer 2 we added to the main-text discussion section E where we discuss the effects of the contacts.
- We have corrected the three “*inaccurate/peculiar statements should be fixed*” pointed out by Reviewer 1.
- In order to follow the guidelines of the Nature communications manuscript checklist we have made the following changes:
 - Rephrased the abstract to make shorter.
 - Changed the colors in figures to color-blind friendly colors (avoided using green and red together in the same figure)
 - Moved the section titled “Experimental” to the end and changed its name to “Methods”.
 - Subheadings were removed from the discussion section.
 - Author contributions and competing interest statements were added.
 - Added the Figure legends again at the end.
 - Added titles to all figures.
 - Removed footnotes from the references (previously we incorporated several footnotes in the references list). The essential footnotes are now incorporated in the text (in the same location where they were referred to in our previous submission).
 - References to the supplemental material were altered to match the guidelines.

Response to the comments of Reviewer 1:

1. The reviewer writes: “*The main text still has presentation issues.... plenty of convincing experimental evidences are given in the SM which appears to be more of a real research paper than the main text... in my opinion the key experimental evidences of the electron overheating bistability should be presented in the main text before Fig. 3. One way of doing so would be to show an example of graphic solution of the thermal balance eq based on the analysis of experimental data (similar to Fig. S7b in the SM) and use it not only to clearly demonstrate the thermal bistability and the magnitude of electron overheating, but also to explain how the minimum and maximum currents I_{c}^{\min} and I_{c}^{\max} are defined, and what are the equal area theorem and the associated current I_p in one sentence... This would also help the reader to understand the physics of switching waves and the lack of hysteresis mentioned in the discussion section without going back and forth between the main text and the SM on each principal issue*”. The referee reiterates “*Instead of showing the redundant and pretty standard dV/dI semilog plots, they should've shown an example of the electron heat balance plot and analyzed three key currents: I_{\min} at which the hot phase can exist, I_{\max} at which the cold SC phase disappears and the minimum propagating current I_p , where $I_{\min} > I_p > I_{\max}$* ”.

Answer: We have followed this advice in the new version of the manuscript and moved the graphical solution (Fig.S7a and Fig.S7b of the previous submission) from the SM to the main text (Fig.3a and Fig.3b). We also added Fig.4d where we show how to extract the minimum propagation current I_p . We believe that this presentation indeed helps the readability of our manuscript.

2. The reviewer writes: “*They should also discuss the effect of contacts and estimate the electron thermal length L to make sure that it is shorter than the length of the sample.*”

Answer: We thank the referee for the suggestion, indeed even at our base temperature of 10mK the electron thermal length is much smaller than the length of the sample $L_T(0.01K) \sim 0.2\mu m$ while our sample is 1mm long. Following the referee’s suggestion we added a discussion of the contacts effect to the main-text.

3. The reviewer added “*Some inaccurate/peculiar statements should be fixed*”:
- a. Page 1: “*Lorentz force acting on the vortices exceeds their binding energy*” – *this sounds like current somehow unbind pairs of vortices, not to mention that a force cannot exceed energy because they have different dimensionalities. Perhaps, something like “the Lorentz force acting on vortices exceeds their depinning force” would be more informative.*

Answer: We corrected this sentence as suggested by the referee.

- b. Page 1: “*near the high critical B* ” – *why cannot the standard “near the upper critical field” be used?*

Answer: We corrected this sentence as suggested by the referee.

- c. *The list of references was cut and pasted from the SI, but Refs. 44-59 were not mentioned in the main text.*

Answer: We thank the referee for bringing that to our attention, in the new version of the manuscript this issue is resolved.

Response to the comment of Reviewer 2:

1. The reviewer writes: “*Contact heating is a very serious concern in critical current measurements. This was well understood more than 50 years ago by one of the pioneers of experimental studies of thin film superconductors, Rolf Glover III, who stated “The second trouble is due to warming of the film by Joule heating which can occur if small normally conducting regions are present, for example where electrical contact is made to the film. The problem is acute since the current densities exceed $10^6 A/cm^2$ ” (Rev. Mod. Phys. 36, 299 (1964).) Indeed, I have personally measured critical currents in thin film superconductors and found that low duty*

cycle pulse measurements can give critical current values that are more than an order of magnitude greater than those of a dc measurement. There are groups all over the world that have the capability to perform pulsed critical current measurements so not having the capability "in house" is not a legitimate excuse. The critical current data in the manuscript must be verified via pulsed measurements or, perhaps by incorporating superconducting contacts, before the authors rush to publication..."

Answer: We agree that contact heating is a serious concern and following the referee's criticism and the suggestion of referee 1 we added a discussion section on the subject. We would like to reiterate that we find it unlikely that the agreement presented in Fig.3c between the measured I_c and I_c^{min} from the heat-balance analysis, which heavily relies on the 4-terminal resistance (measured at $I \ll I_c$), is coincidental. We are also not convinced that the referee's suggested heating mechanism would result in the large magnetic field dependence of I_c as displayed in Fig.1c. We would also like to point out that the largest critical current densities we report in this work are of the order of 200A/cm², which is almost 4 orders of magnitude lower than the results of Rolf Glover III which the referee cites above. Lastly, we would like to draw the referee's attention to figures 2b and 3 of reference [1] where the authors measured a:InO samples of very similar transport properties to those presented in our work, using the same methods we used, with the same type of contacts as ours (in fact these samples were prepared by us in our lab) and measure critical currents that are consistent with ours critical currents. In these figures the authors compare J_c 's of three samples where for two of these samples, ITb1 and J038, J_c is almost identical. On the other hand, the dimensions of these samples and specifically the overlap area between the a:InO and Ti/Au contacts (a:InO was deposited on top of the contacts) is different between the two samples where the overlap for sample J038 (200 μ m on 100 μ m) is ~25 times larger than for sample ITb1 (40 μ m on 20 μ m). If the critical current is indeed due to the contacts, as the referee claims, such a difference in the contact overlap should have resulted in a significant difference in Joule-heating at the contacts which, as plotted in these figures, is not the case.

References

- [1] B. Sacépé, J. Seidemann, F. Gay, K. Davenport, A. Rogachev, M. Ovadia, K. Michaeli and M. V. Feigel'man, "Low-temperature anomaly in disordered superconductors near B c2 as a vortex-glass propert," *Nature physics*, vol. 15, p. 48, 2019.